# Magnetic Resonance Imaging, Computed Tomographic and Radiographic Findings in the Metacarpophalangeal Joints of 31 Warmblood Showjumpers in Full Work and Competing Regularly

**DOI:** 10.3390/ani14101417

**Published:** 2024-05-09

**Authors:** Annamaria Nagy, Sue Dyson

**Affiliations:** 1Department and Clinic of Equine Medicine, University of Veterinary Medicine Budapest, Dóra Major, 2225 Üllő, Hungary; 2The Cottage, Church Road, Market Weston, Diss IP22 2NX, UK; sue.dyson@aol.com

**Keywords:** sports horse, fetlock, MRI, CT

## Abstract

**Simple Summary:**

Magnetic resonance imaging (MRI) and computed tomography are increasingly used in the diagnosis of fetlock region pain in sports horses and racehorses. To establish an accurate diagnosis, it is paramount to be familiar with abnormalities that can be seen in horses in full work and without relevant lameness. The aim of this study was to describe MRI, computed tomographic (CT) and radiographic findings in the front fetlocks of showjumpers who were in full training and competing regularly. Thirty-one horses without previous fetlock disease or current lameness were included. The most common finding was densification (bone hardening) in the region just under the joint surface in the cannon bone and long pastern bone. These changes likely reflect an adaptive response to exercise. Small indentations and focal decrease or loss of bone density were seen in multiple locations and might reflect early stages of disease or might be incidental. In conclusion, MRI and CT abnormalities previously associated with lameness were seen in the front fetlocks of showjumpers in full work and without relevant lameness. This should be considered when interpreting imaging studies of lame horses.

**Abstract:**

There is a limited description of magnetic resonance imaging (MRI) and no information on computed tomographic (CT) findings in the fetlock of non-lame, non-racing sports horses. This study aimed to document comparative CT, MRI and radiographic findings in the metacarpophalangeal joints of showjumpers in full work. Clinical and gait assessments, low-field MRI, fan-beam CT and radiographic examinations of both metacarpophalangeal joints were performed on 31 showjumpers. Images were analysed descriptively. In most limbs (53/62, 85.5%), there were CT and MRI changes consistent with densification in the sagittal ridge and/or condyles of the third metacarpal bone (McIII). Hypoattenuation (subchondral bone resorption) was seen in CT reconstructions in the metacarpal condyle dorsoproximally (n = 2) and dorsodistally (n = 1), in the sagittal groove (n = 2) and medial fovea (n = 1) of the proximal phalanx. The McIII resorptive lesions were detected on MR images but not the proximal phalanx lesions. None were identified on radiographs. In conclusion, MRI and CT abnormalities previously associated with lameness were seen in the front fetlocks of showjumpers without relevant lameness. Densification in the sagittal ridge and the metacarpal condyles likely reflects an adaptive change to exercise. Subchondral bone resorption may indicate an early stage of disease; follow-up information is needed to establish its clinical significance.

## 1. Introduction

Magnetic resonance imaging (MRI) and computed tomography are being used with increasing frequency in the diagnosis of fetlock region pain in both sports horses and racehorses [1,2,3,4,5,6,7,8,9,10,11,12,13,14,15,16,17]. For accurate interpretation of imaging studies of lame horses, knowledge of anatomical variations is essential, and information on abnormalities that can be seen without clinical significance in horses of various ages, work disciplines and exercise levels would be desirable. Description of MRI findings in non-lame sports horses are limited to the foot [18], proximal metacarpal region [19,20,21,22], carpus [21], proximal metatarsal region and tarsus [23,24]. Magnetic resonance abnormalities in the contralateral non-lame limb of unilaterally lame horses have been reported in the metacarpophalangeal [17,25] and proximal metacarpal regions [26]. Comparative MRI and CT studies in the fetlock are mostly limited to cadaver studies and racehorses [27,28,29,30,31,32,33]. Recently, comparative low-field MRI and cone-beam CT findings in the metacarpophalangeal and metatarsophalangeal joints in standing sedated horses were described [16]. This paper focussed on the comparison of the imaging modalities, and no signalment or other clinical information about the horses was provided. Magnetic resonance imaging, computed tomographic (CT) and radiographic findings in the metacarpophalangeal joint of non-lame Thoroughbred yearlings have recently been documented [34]. These results are likely not applicable to older sports horses that have been exposed to exercise for several years.

To date, there is no description of MRI and CT findings in the fetlock of non-lame, non-racing sports horses. The aim of this study was to document subjective comparative fan-beam CT, low-field MRI and radiographic findings in the metacarpophalangeal joints of showjumpers without relevant lameness, in full work, and competing regularly.

## 2. Materials and Methods

This was a descriptive, prospective cross-sectional study, with a convenient population of horses in full training and competing regularly in showjumping. Horses were eligible to participate if they were considered to be free from relevant lameness, had no history of metacarpophalangeal joint disease or treatment and fell into one of the three age and competition level categories: (1). 5–6 years old, competing at ≥110 cm; (2). 7–9 years old, competing at ≥120 cm; and (3). >9 years old, competing at ≥130 cm. The recruitment phase was set at 12 months; during this time, 31 eligible horses were examined. Prior to accepting horses for an assessment, each horse’s home veterinary surgeon was interviewed to minimise recall bias of the owner with regards to lameness, metacarpophalangeal joint disease and treatment.

A detailed clinical history was obtained, including the level of competition and training, any previous lameness or joint medication and the recent work intensity. Clinical examination and lameness assessment were performed, including subjective and objective (Equinosis Q Lameness Locator^®^, Columbia, MO, USA) gait analysis in a straight line in trot on a firm surface, distal forelimb flexion tests (one-minute duration), and subjective evaluation on the lunge on soft and hard surfaces on both the left and right reins. Horses were included if they did not show overt consistent forelimb lameness (grade > 1/8) [35] under any circumstances. A horse with a consistent lameness grade > 1/8, seen under any circumstance (in hand in straight lines, on the lunge on either soft or hard surfaces), would have been excluded.

Diagnostic imaging of both metacarpophalangeal joints was performed in a standing position under sedation, using a combination of romifidine (Boehringer Ingelheim, Ingelheim, Germany) (0.01 mg/kg IV), detomidine (Orion Pharma, Budapest, Hungary) (0.01 mg/kg IV) and butorphanol (Bioveta, 68323 Ivanovice na Hané, Czech Republic) (0.02 mg/kg IV). Low-field MRI was performed in a 0.27 T open magnet (Hallmarq Veterinary Imaging Ltd., Guildford, UK) using a clinical protocol: T1-weighted and T2*-weighted gradient echo and short-tau inversion recovery sequences in sagittal, frontal and transverse planes, and T2-weighted fast spin echo sequences in a transverse plane (Table 1). Transverse plane sequences were set perpendicular to the long axis of the proximal phalanx. Additional T1-weighted gradient echo and T2-weighted fast spin echo transverse sequences of the most distal aspect of the suspensory ligament branches were also acquired, set perpendicular to the long axis of the third metacarpal bone.

Fan-beam CT examination was performed using a system customised for the examination of horses in a standing position (Qalibra CT System, Vet-DICon, Zossen, Germany), equipped with a 16-detector multislice helical scanner with a 90 cm bore diameter (Canon Aquilion LB, Tokyo, Japan). Settings for the acquisition protocol included 0.5 mm slice thickness, 135 kV and 350 ms, 320 mm field of view and 0.5 s rotation time. Limbs were scanned from the distal third of the metacarpal region to the level of the proximal interphalangeal joint. Radiographic examination included lateromedial, dorsal 10° proximal—palmarodistal oblique, dorsal 45° lateral–palmaromedial and palmar 45° lateral–dorsomedial oblique, and additional (lateromedial (flexed), dorsopalmar (flexed), dorsal 70° proximal 45° lateral–palmarodistomedial oblique and dorsal 70° proximal 45° medial–palmarodistolateral oblique and dorsoproximal–dorsodistal (flexed) (‘skyline’)) views (Fuji DR-ID1270, Fujifilm Corporation, Tokyo, Japan). All diagnostic imaging acquisitions were performed within 30 h.

Images were assessed subjectively in medical viewing software (JiveX DICOM Viewer Diagnostic Advanced 5.2^®^, Visus Health IT GmbH, Bochum, Germany) following a protocol adapted to each imaging modality [34]. The presence and location of abnormalities in the cortical, subchondral, trabecular and compact bone, as well as in soft tissues, were recorded. Conclusions were reached by consensus of the authors (a Diplomate of the American and European Colleges of Veterinary Sports Medicine and Rehabilitation and an Associate of the European College of Veterinary Diagnostic Imaging). First, all computed tomographic, then MRI and radiographic studies were assessed in chronological order of acquisition. If there was a discrepancy between specific imaging findings among modalities, images were re-assessed, and a final conclusion was made.

For ease of reading, concurrent hyperattenuation in CT reconstructions and hypointense signal in T1 and T2*-weighted gradient echo and T2-weighted fast spin-echo MR images in the trabecular bone will be referred to as ‘densification’. The extent of densification in the metacarpal condyles and the sagittal ridge was established in transverse MR images and CT reconstructions and was graded as mild if it extended up to one-quarter of the dorsopalmar depth, moderate if it extended from one-quarter to half of the dorsopalmar depth, and extensive if it extended beyond half of the dorsopalmar depth. The subchondral bone plate of the proximal phalanx was assessed for mediolateral symmetry on frontal CT reconstructions in the middle, central and palmar thirds and described as symmetrical, thicker medially or thicker laterally. The pattern of dorsopalmar thickness was evaluated subjectively in sagittal reconstructions and described as even or thicker dorsally, in the middle or in the palmar third. The sagittal groove and the regions of the medial and lateral foveae articularis were each assessed independently. No measurements were acquired; objective image analysis will be performed in a separate study.

## 3. Results

All 31 horses included in the study were Warmbloods; there were 14 mares, 12 geldings and 5 stallions. Eight horses were competing at 110–115 cm level (mean: 111 cm, mean age 5.9 years), nine horses at 120–125 cm level (mean: 123 cm, mean age 6.7 years) and 14 horses at >130 cm level (mean: 137 cm, mean age 10.4 years). The horses’ mean body weight was 570.5 ± 45.5 kg and their mean height was 168.3 ± 4.4 cm. The metacarpophalangeal joint was distended in 16 limbs of 10 horses; the distension was moderate in two and minimal to mild in the remaining limbs. There was a restricted range of motion in five fetlocks of three horses; no horses showed a pain response during passive manipulation. Subjectively, none of the horses showed overt consistent forelimb lameness under any circumstances. When trotted in a straight line, subtle (<1/8) or intermittent forelimb lameness was noted in four horses (two left fore and two right forelimb lameness), and four horses showed subtle (≤1/8) right hindlimb lameness. On objective assessment, 19 horses showed head movement asymmetry on the right forelimb (mean vector sum 5.8 ± 3.6 mm) and 12 horses on the left forelimb (mean vector sum 6.7 ± 2.9 mm).

Distal limb flexion tests of the forelimbs were slightly positive (mild, grade 2/8 lameness for a few strides) in seven limbs of six horses. When lunged on a soft surface, two horses showed subtle (<1/8) right hindlimb lameness and one horse showed subtle left forelimb lameness. On the hard surface of the lunge, five horses showed subtle right forelimb lameness; one also showed subtle left forelimb lameness and another showed subtle right hindlimb lameness.

### 3.1. Third Metacarpal Bone

Thickening of the subchondral bone and/or densification of the trabecular bone of the sagittal ridge of the third metacarpal bone was detected dorsally in 51/62 (82.3%) limbs in MR images and 52/62 (83.9%) limbs in CT images, and in the palmar aspect in 26/62 (41.9%) limbs in MR and CT images (Figure 1 and Figure 2). A cone-shaped hypointense signal in the dorsal aspect of the sagittal ridge was seen in 3 limbs (of horses aged 5, 6 and 7 years) in MR images only. A hypoattenuating lesion in the dorsodistal aspect of the sagittal ridge was seen in 11/62 (17.7%) limbs in CT images; 4 of the 11 lesions were identified on MR images as focal hyperintense signals in all image sequences, and 2/11 were detected as radiolucent lesions in lateromedial (flexed) and ‘skyline’ radiographs (Figure 2). In two of the three limbs with cone-shaped hypointense signals on MR images, there were small hypoattenuating lesions in the dorsodistal aspect of the sagittal ridge in CT reconstructions, one of which was also detected on MR images as focal hyperintensity.

In most limbs, densification was seen in both dorsal and palmar aspects of the medial condyle (diffuse hypointense signal in 48/62 (77.4%) and hyperattenuation in 51/62 (82.3%) limbs) (Figure 2). In the dorsal aspect of the medial condyle, densification was detected more frequently in CT images (53/62 limbs, 85.5%) than in MR images (24/62 limbs, 38.7%), while in its palmar aspect, detection of densification was similar in the two modalities (MRI 24/62, 38.7%, CT 25/62, 40.3%). In the lateral condyle, densification was identified in the palmar aspect in 53/62 (85.5%) limbs in MR images and in 55/62 (88.7%) limbs in CT reconstructions. Densification was seen less frequently in the dorsal aspect of the lateral condyle (34/62 (54.8%) in MR images and in 53/62 (85.5%) limbs in CT reconstructions). The pattern was usually bilaterally symmetrical. In all but one horse, densification was considered mild to moderate. Extensive densification, affecting the entire dorsopalmar depth of both condyles, was seen in only one horse competing at 140 cm.

There was a hypoattenuating lesion in the dorsal half of the medial condyle of the third metacarpal bone in five limbs of four horses, one in the dorsal aspect of the weight-bearing surface (Figure 3) and three in the proximal aspect of the condyle (Figure 4). The distal lesion and one of the proximal lesions were also detected on MR images as focal hyperintensity in all image sequences, surrounded by a rim of a hypointense signal. In three limbs of two horses, there was a smooth indentation in the medial parasagittal groove, seen only in CT reconstructions. None of these condylar lesions were detected radiographically. In one limb, there was a faint linear hypoattenuating lesion in the subchondral bone of the palmar aspect of the medial condyle, only detectable on CT reconstructions (Figure 5). Periarticular modelling was identified on both the medial and the lateral margins of the third metacarpal bone in three limbs of two horses in CT and MR images (Figure 2).

### 3.2. Proximal Phalanx

Thickening of the proximal subchondral bone of the proximal phalanx was seen in all but one limb in CT and MR images (Figure 2, Figure 3, Figure 4, Figure 6 and Figure 7). Asymmetrical thickness was seen in the dorsal third of the bone in 8/62 (12.9%) limbs (thicker medially in all), in 18/62 (29.0%) limbs in the central third of the bone (thicker medially in 16 and laterally in 2 limbs) and in the palmar third in 35/62 (56.5%) limbs (thicker medially in 25 and laterally in 10 limbs). The dorsopalmar pattern of subchondral bone thickness had the following distribution: in most limbs in both the medial and the lateral fovea, the subchondral bone was thickest in the palmar third (medial 59/62, 95.2%, lateral 57/62, 91.9%). In the sagittal groove, in the majority of limbs, the subchondral bone was thickest in the middle third (35/62, 56.5%); in 17 limbs, it was even from dorsal to palmar, and in 10 limbs, it was thickest in the dorsal third.

In both limbs of one horse, there was a shallow defect (indentation) in the sagittal groove, surrounded by mildly irregular focal hypoattenuation and thickening of the subchondral bone. In one of the limbs, there was an associated mild hyperintense signal in T1- and T2*-weighted GRE images (Figure 6); in the other limb, only a small indentation could be identified in MR sequences. In two additional limbs, there were subtle hypoattenuations in the sagittal groove in CT reconstructions, not associated with detectable MRI abnormalities. In one limb, there was a small hypoattenuating lesion in the medial aspect of the sagittal groove detectable only in the CT reconstruction, surrounded by marked thickening of the subchondral bone evident in both the CT and MR images (Figure 7). None of these findings were identified radiographically. In one limb, there was focal hypoattenuation in the medial fovea of the proximal phalanx, which was not detected on MR images or radiography.

Periarticular modelling of the proximal phalanx was identified in 26/62 (41.9%) limbs in CT images (on both the medial and lateral aspects in 23/26 limbs and on the lateral aspect only in 3 limbs), in 11/62 (17.7%) limbs in MR images (on both medial and lateral aspects in 4 limbs, on the medial aspect only in 1 limb, on the lateral aspect only in 6 limbs (Figure 2 and Figure 7), and in 7/62 (11.3%) limbs (on both the medial and lateral aspects) on radiographs. There was mild modelling of the dorsoproximal aspect of the proximal phalanx on CT reconstructions and in radiographs in 7/62 (11.3%) limbs.

### 3.3. Proximal Sesamoid Bones

There was a modelling of the dorsoproximal and/or dorsodistal aspect of the medial proximal sesamoid bone in 25/62 (40.3%) limbs and a modelling of the lateral proximal sesamoid bone in 22/62 (35.5%) limbs in CT reconstructions (Figure 5). A small focal hyperattenuation was consistent with an apical fragment of both proximal sesamoid bones or mineralisation in the insertion of the suspensory ligament, which was seen in CT images in one limb. In another limb, there was focal hyperattenuation in the intersesamoidean region.

### 3.4. Soft Tissues

No significant soft tissue abnormalities were detected in either MR images or computed tomographic reconstructions.

## 4. Discussion

This is the first report of comparative MRI, CT and radiographic findings in the metacarpophalangeal joint of Warmblood showjumpers in full work and competing regularly. The current study is the first part of a longitudinal project; horses are followed for two to three years with clinical examination and diagnostic imaging repeated at intervals of approximately eight months.

All horses underwent detailed clinical examination, subjective and objective gait evaluation, and flexion tests. None showed a baseline lameness or exacerbation of lameness following flexion of the distal limb or lungeing, which was considered clinically significant by the clinician performing these evaluations (Diplomate of the American and European Colleges of Veterinary Sports Medicine and Rehabilitation, over two decades of experience in lameness investigation in sports horses). Subjective gait evaluation revealed subtle forelimb gait abnormality in a straight line and/or on the lunge in a small proportion of horses (7/31, 22.6%). The maximum grade of baseline lameness was 1/8 (just perceptible), and following the distal limb flexion test, it was 2/8 [35]. A subtle gait abnormality is often seen in competition horses and frequently does not limit performance [36,37]. In a recent study of showjumpers, dressage horses and eventers competing at an international level, 88% were lame based on subjective evaluation and 89% based on objective evaluation. No significant correlation was seen between lameness and performance parameters [36]. In the current study, the vector sum of head motion asymmetry was higher in some horses than the threshold for lameness detection recommended for this objective lameness assessment tool using body-mounted inertial sensors (8.5 mm) [38]. Head motion asymmetry exceeding the published threshold level has been documented in event horses competing at 3* and 4* levels [37] and in young Warmblood horses in full work [39]. Mild movement asymmetry may be related to laterality [38,39,40] rather than pain-induced lameness. In the current study, due to time constraints, horses were not assessed ridden; lameness manifesting only during ridden exercise and/or behavioural signs associated with musculoskeletal pain may have been missed. However, more detailed clinical and gait evaluation findings are reported for this study population than in most publications describing advanced diagnostic imaging of non-lame limbs of horses [18,19,20,21,23,24,25,26]. Moreover, investigation of the prevalence of potential musculoskeletal problems was not within the scope of this study.

The most common MRI and CT finding, seen in over 80% of limbs, was densification in the trabecular bone of the sagittal ridge and the condyles of the third metacarpal bone, which likely reflects osseous adaptive change in response to exercise [41,42]. In non-lame Thoroughbred yearlings that had been exposed to no or minimal training, mild densification (mild diffuse hyperattenuation in CT reconstructions) was seen in the dorsal aspect of the medial condyle in 33/80 (41.3%) limbs and in the palmar aspect of the lateral condyle in 25/80 (31.3%) limbs [34], which can be related to bone modelling due to exercise early in life [34,42,43]. In the current study population of showjumpers, typically, the most prominent densification was seen in both the dorsal and palmar aspects of the medial condyle and in the palmar aspect of the lateral condyle. For a valid comparison of the densification pattern between work disciplines, data from non-lame older Thoroughbred racehorses would be required. The authors acknowledge that in some horses, differentiation between the thickening of the subchondral bone and densification of the adjacent trabecular bone can be challenging. In vitro biomechanical studies on pressure distribution on the proximal phalanx and biomechanical properties of the articular cartilage in the metacarpophalangeal joint have not demonstrated a mediolateral difference [44,45]. In vitro limbs are loaded along the limb axis, while in vivo loading is likely more asymmetrical. However, another in vitro study identified greater principal strain in the medial than the lateral aspect of the proximal phalanx but no asymmetry in the metacarpal condyles [46]. To our knowledge, no specific studies have investigated the loading pattern of the third metacarpal bone in showjumpers.

In the palmar aspect of the condyles, densification was identified more frequently in CT than in MR images. The reason for this is not clear. It is possible that focal low-grade trabecular hypointensity is missed due to volume averaging and the normal trabecular bone signal intensity dominates within the thickness of a 5 mm slice.

Subchondral bone thickening in the proximal phalanx was seen in most horses and is likely to reflect osseous modelling as an adaptive change to exercise [43,47]. When asymmetry was seen in the majority of limbs, the subchondral bone plate was thicker in the medial than in the lateral fovea. A greater cartilage degeneration index was seen in the medial than in the lateral fovea in an in vitro study of horses of unknown history [48]. This probably indicates asymmetrical loading of the proximal aspect of the proximal phalanx in vivo. Mediolateral asymmetry and dorsopalmar thickness pattern were assessed in frontal or sagittal CT reconstructions and not on MRI studies to avoid potential misinterpretation/misclassification arising from the greater slight thickness and imperfect (i.e., not tangential) positioning resulting in some obliquity of frontal and sagittal MR images.

Densification in the sagittal ridge and metacarpal condyles, similar to that seen in the current study, has been described as a significant finding (and also referred to as ‘abnormal mineralisation’) in MRI reports of horses with fetlock region pain [1,5,6,17]. Trabecular bone densification can be associated with pain, potentially because of increased intraosseous pressure [47], and it is not possible to conclude that the abnormalities seen in this population will not progress and contribute to lameness. However, the high prevalence seen in the current study suggests that the clinical significance of sagittal ridge and condylar densification should be assessed with caution.

The small lytic/resorptive lesions in the dorsodistal aspect of the sagittal ridge of the third metacarpal bone are consistent with osteochondrosis [49]. Similar lesions were detected in CT and MR images of non-lame Thoroughbred yearlings [34] and on radiographs of two lame Warmbloods aged 14 months and 7 years [49]. In the study including the two Warmblood horses, lameness had been localised to the fetlock by diagnostic analgesia or the presence of distension of the metacarpophalangeal joint. The exclusion of other lesions potentially causing or contributing to the lameness was not discussed. In immature Thoroughbreds [34], the focal lesions were surrounded by much more extensive densification than in the mature showjumpers in the current study. Cone-shaped densification was common in Thoroughbred yearlings [34], occurring in 21.3% of limbs in association with lesions consistent with osteochondrosis of the sagittal ridge, but was seen in only 3/62 (4.8%) limbs of three horses in the current study. In all three showjumpers, cone-shape densification (mineralisation) was only detected in MR but not in CT images. This is consistent with the young Thoroughbred study, where only 26% of cone-shaped densification seen on MR images was detected in CT reconstructions [34]. It was suggested that in MR images, signal intensity is averaged across 5 mm and results in more obvious changes than in CT reconstructions that are based on 0.5 mm thick slices [34]. The three showjumpers were relatively young (5–7 years) compared with the mean age of all horses in the study (10.4 years). It is possible that the severity and/or extent of trabecular bone densification decreases as the bone remodels [50]. The effect of exercise on bone structure has been investigated extensively in Thoroughbred racehorses and immature Warmblood horses [51,52,53,54,55,56]. To our knowledge, investigation of the effect of exercise on the musculoskeletal system of mature non-racing sports horses is limited to a case series of endurance horses, examined twice with an interval of approximately six months, in which MRI findings in the proximal metacarpal region were described [22].

A focal subchondral bone lesion in the dorsoproximal aspect of the medial condyle of the third metacarpal bone has been associated with lameness in an MRI study [5]. The bilateral presence of the lesion in a non-lame horse in the current study suggests a developmental abnormality and may reflect an uncommon form of osteochondrosis [57]. The authors have seen identical lesions in two additional non-lame horses. A subchondral bone lesion in the dorsodistal (weight-bearing) aspect of a condyle of the third metacarpal bone is more likely to be associated with clinically significant pathology [1,5]. A subchondral bone lesion in the palmar aspect of the condyle may represent an early stage of palmar osteochondral disease [12].

Magnetic resonance imaging and/or CT abnormalities associated with focal demineralisation were only seen in a few horses. Shallow defects (indentation) in the subchondral bone in the sagittal groove and medial or lateral fovea of the proximal phalanx were described in young non-lame Thoroughbreds [34]. In other investigations, similar findings were referred to as either fissures in a cadaver CT and MRI study of racehorse fetlocks [33], minor shallow defect (≤1 mm in depth) or microfissures (≤3 mm in length) in an in vivo MRI study of sports horses [17]. In agreement with a previous publication [17], the authors of the current study also propose that these smooth indentations in the subchondral bone are likely an anatomical variation and are of developmental origin. ‘Fissure’ suggests a pathological process, and, therefore, the authors prefer the terms indentation or shallow defect. Disease or injury of the sagittal groove of the proximal phalanx is an important cause of lameness in sports horses and can progress into fractures [6,17,58]. Recently, sagittal groove lesions were documented in non-lame limbs in an MRI study [17]. Longitudinal studies are needed to establish any potential association between a smooth indentation of the articular surface, as seen in the current study, and subchondral bone disease. Further investigation is also warranted to evaluate whether focal hyperintensity/hypoattenuation distal to the indentation increases the risk of clinically significant disease.

Diffuse STIR hyperintensity has been described in the sagittal ridge and the dorsal aspect of the medial condyle of the third metacarpal bone in non-lame limbs [25]. This finding was not observed in the current study. This may be related to different populations. The authors of the current manuscript were rather conservative in their judgement of abnormalities, and hyperintense STIR signal was only noted if any possibility of movement or flow artefact was excluded. Focal STIR hyperintensity was seen in some horses with focal subchondral demineralisation. Follow-up studies are needed to establish whether these will progress into lesions associated with lameness.

Mild periarticular modelling was common in MRI and CT images but not in radiographs. This likely reflects low-grade and/or subclinical osteoarthritis that was not currently causing pain or reduced performance.

Soft tissue abnormalities were the most common lesions in several previous MRI studies in horses with fetlock region pain [5,15]. In the current study, significant soft tissue lesions were not detected with any imaging modality. This may suggest that most soft tissue abnormalities seen in lame horses are likely to be clinically significant. It is also possible that mild changes (e.g., focal hyperintensity in the oblique sesamoidean ligaments without enlargement of cross-sectional area) were not considered significant by the current authors, and they might have been judged differently by others. In one limb, there was focal hyperattenuation in the region of the intersesamoidean ligament, indicating mineralisation and in another limb, focal hyperattenuation was seen at the apex of the proximal sesamoid bone. Neither lesion was detected on radiography or MRI. Mineralisation in soft tissues can be seen as an incidental finding [59]. Heterotopic mineralisation in locations similar to those seen in the current study was described on CT and MR images in a recent equine fetlock cadaver study without indication of potential clinical significance; no clinical information on the horses was provided [31]. In the current study, no associated soft tissue abnormalities were identified.

The main limitation of the study is that although owners declared no knowledge of fetlock region problems, previous metacarpophalangeal joint disease (e.g., in previous ownership) cannot be excluded in all horses. Moreover, the sample population was small; therefore, no statistical comparisons among groups were performed. Images were not assessed blindly and independently; the purpose of the study was not to blindly evaluate each imaging modality and compare the results, but to present a subjective assessment of what was present, as performed in a clinical situation. Although objective comparison of imaging modalities was beyond the scope of this study, the authors acknowledge the difficulty in creating precisely ‘matching’ transverse CT and MR images because of the different positions of the distal limb in the two image acquisition systems.

## 5. Conclusions

Magnetic resonance imaging and CT abnormalities previously associated with lameness can be seen in the metacarpophalangeal joints of non-lame Warmblood showjumpers. Densification in the sagittal ridge and the metacarpal condyles and thickening of the subchondral bone of the proximal phalanx were seen in most horses. These likely reflect adaptive changes, and their clinical significance should be interpreted with caution by lame showjumpers. Subchondral bone lesions may reflect the early stages of disease; follow-up information is needed to establish their long-term clinical significance.

## Figures and Tables

**Figure 1 animals-14-01417-f001:**
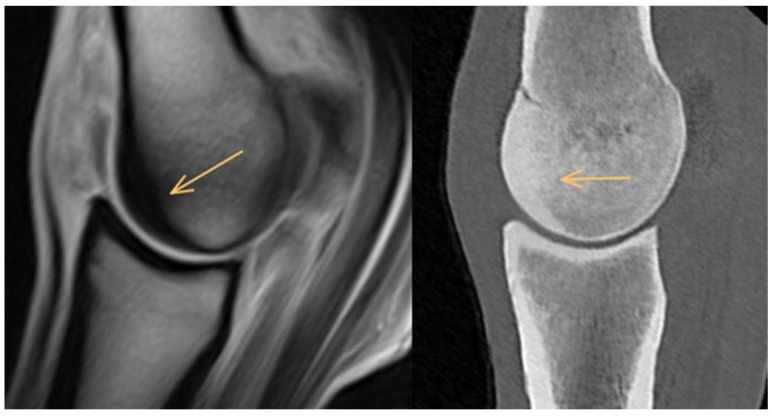
Sagittal T1-weighted gradient echo magnetic resonance image and computed tomographic reconstruction of a metacarpophalangeal joint of an 8-year-old showjumper competing at 125 cm. Dorsal is to the left. There is thickening of the subchondral bone and mild diffuse hypointense signal/patchy hyperattenuation (arrows) in the dorsal half of the trabecular bone of the sagittal ridge of the third metacarpal bone.

**Figure 2 animals-14-01417-f002:**
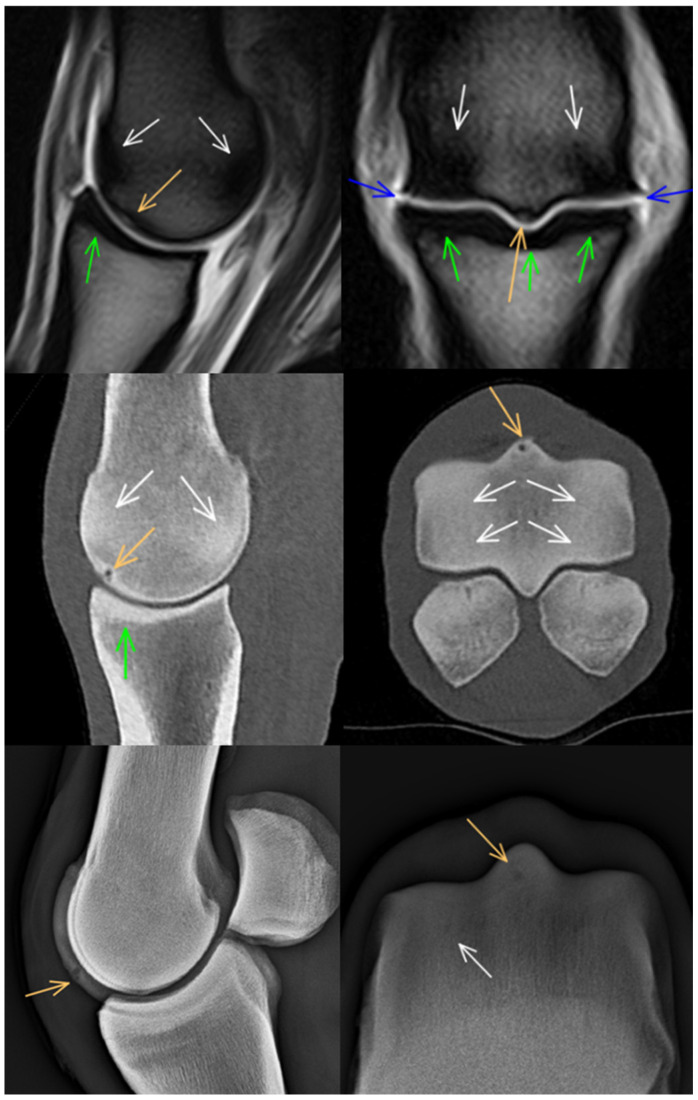
Sagittal and frontal T2*-weighted gradient echo magnetic resonance images, sagittal and transverse computed tomographic reconstructions and lateromedial (flexed) and dorsoproximal-dorsodistal (flexed) radiographs of a metacarpophalangeal joint of a 13-year-old showjumper competing at 140 cm. Dorsal and medial are to the left. There is focal hyperintensity/hypoattenuation/lucency in the dorsodistal aspect of the sagittal ridge of the third metacarpal bone (orange arrows). Note also the changes consistent with densification in the dorsal and palmar aspects of the sagittal ridge and the medial and lateral condyles (white arrows) and the mildly thickened subchondral bone in the dorsal half of the proximal phalanx (green arrows). There is periarticular modelling on the distal medial and lateral aspects of the third metacarpal bone and, to a lesser extent, on the medial and lateral aspects of the proximal phalanx (blue arrows).

**Figure 3 animals-14-01417-f003:**
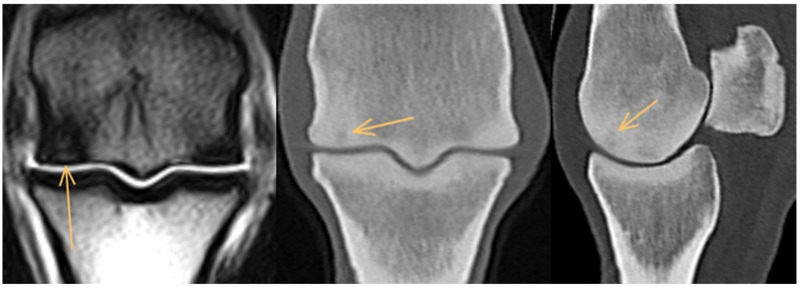
Frontal T2*-weighted gradient echo magnetic resonance image and frontal and parasagittal computed tomographic reconstructions of a metacarpophalangeal joint of a 6-year-old showjumper competing at 120 cm. Medial and dorsal are to the left. There is focal hyperintense signal/hypoattenuation in the dorsodistal aspect of the medial condyle (arrows), surrounded by a zone of reduced signal intensity/hyperattenuation consistent with an increase in bone density.

**Figure 4 animals-14-01417-f004:**
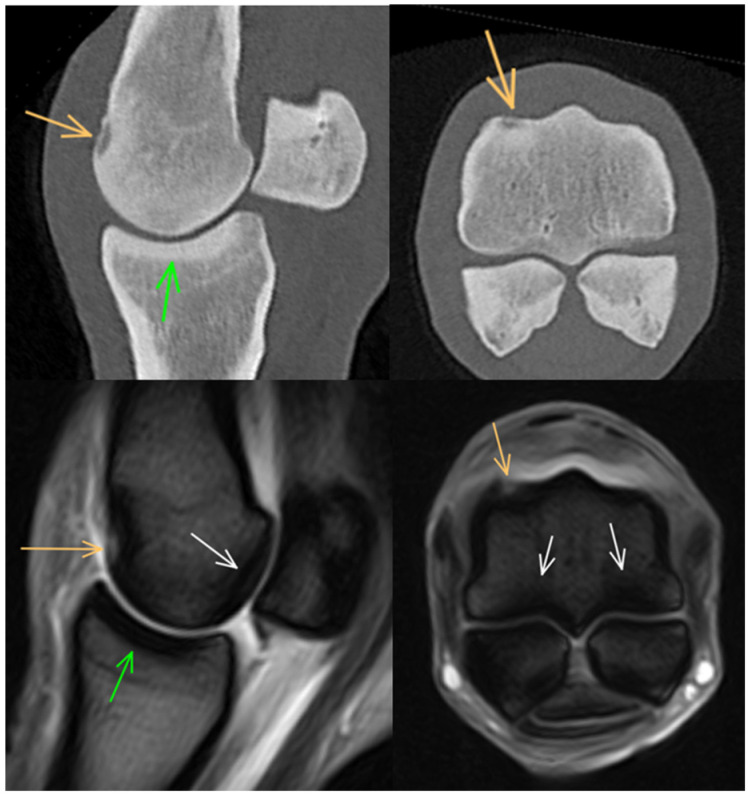
Parasagittal and transverse computed tomographic reconstructions and T2*-weighted gradient echo magnetic resonance images of a metacarpophalangeal joint of a 6-year-old showjumper competing at 110 cm. Dorsal and medial are to the left. There is hypoattenuation/focal hyperintense signal in the dorsoproximal aspect of the medial condyle, surrounded by a rim of hyperattenuation/low signal intensity consistent with an increase in bone density (orange arrows). There is mild hypointense signal in the palmar aspect of the metacarpal condyles (white arrows) and mild thickening of the subchondral bone of the proximal phalanx (green arrows).

**Figure 5 animals-14-01417-f005:**
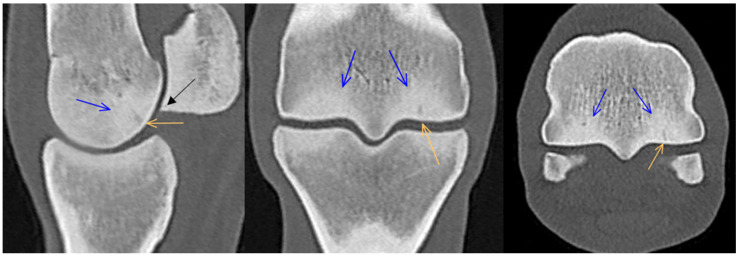
Parasagittal, frontal and transverse computed tomographic reconstructions of a metacarpophalangeal joint of an 8-year-old horse competing at 125 cm. Medial and dorsal are to the left. There is focal linear hypoattenuation in the palmar aspect of the lateral condyle of the third metacarpal bone (orange arrows). Note the moderate diffuse hyperattenuation in the palmar aspect of both condyles (blue arrows) and mild modelling at the dorsodistal aspect of the lateral proximal sesamoid bone (black arrow).

**Figure 6 animals-14-01417-f006:**
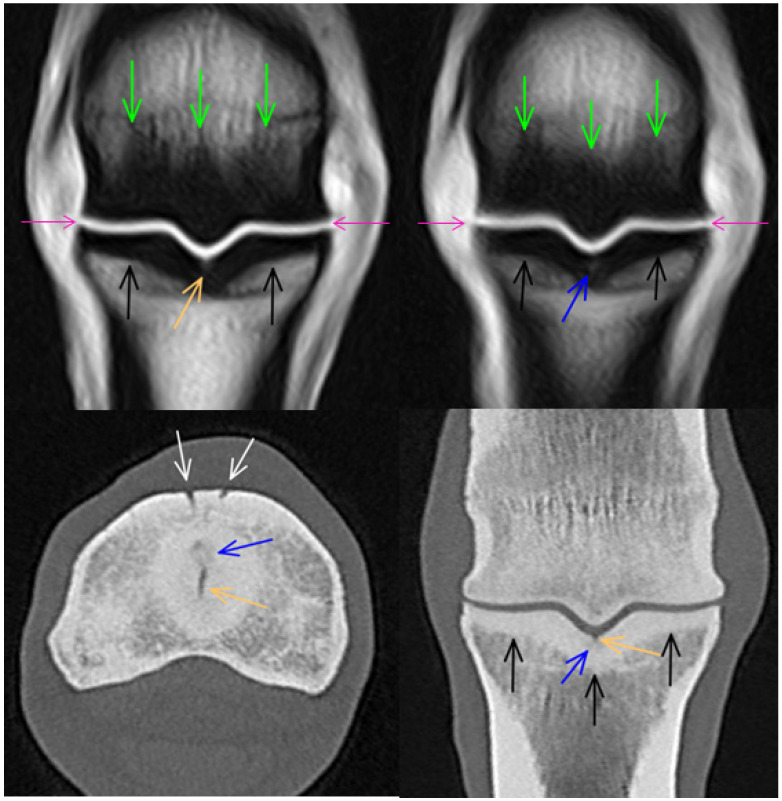
Frontal T1-weighted gradient echo magnetic resonance images (right image acquired just dorsal to the left image) and transverse and frontal computed tomographic (CT) reconstructions of a 14-year-old showjumper competing at 145 cm. There is an indentation/shallow defect in the sagittal groove of the proximal phalanx (orange arrows), surrounded by mild hyperintense signal/hypoattenuation. Just dorsal to the middle of the weight-bearing surface, there is an ill-defined area of hypoattenuation (blue arrows) in the sagittal groove. The subchondral bone is diffusely thickened along the proximal articular surface of the proximal phalanx (black arrows). Note the hypointense signal consistent with densification in the sagittal ridge and both condyles of the third metacarpal bone (green arrows). These were graded as extensive on transverse images. This is less obvious on CT images, due to the different positions of the third metacarpal bone in relation to the proximal phalanx during image acquisition of the two modalities. There is mild periarticular modelling on the medial and lateral aspects of the proximal phalanx and the third metacarpal bone, which is more prominent on magnetic resonance images than on CT reconstructions (purple arrows). There are prominent vascular channels in the dorsoproximal aspect of the proximal phalanx (white arrows).

**Figure 7 animals-14-01417-f007:**
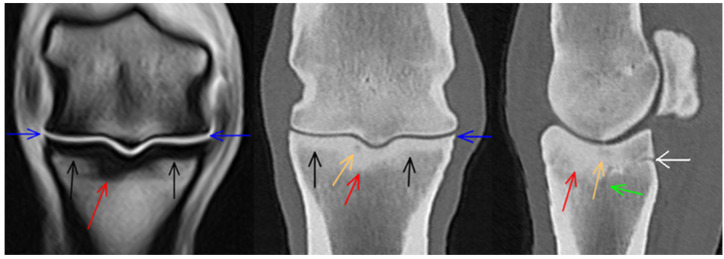
Frontal T1-weighted gradient echo magnetic resonance image and computed tomographic (CT) reconstruction of a metacarpophalangeal joint of a 7-year-old showjumper competing at 125 cm. Medial and dorsal are to the left. There is focal marked thickening of the subchondral bone and/or densification of the trabecular bone in the medial aspect of the sagittal groove of the proximal phalanx (red arrows) and a small hypoattenuating lesion (orange arrow), which does not appear to communicate with the articular surface. There is an obliquely linear hypoattenuation distal to the subchondral lesion, likely consistent with an abnormal vascular channel (green arrow). The remaining subchondral bone plate is mildly thickened (black arrows). There is mild periarticular modelling on the medial and lateral aspects of the proximal phalanx, more prominent on magnetic resonance images than on CT reconstructions (blue arrows). Note the enlarged vascular channel in the palmar aspect of the proximal phalanx in the parasagittal CT reconstruction (white arrow).

**Table 1 animals-14-01417-t001:** Pulse sequence parameters used in a 0.27 T magnet to image the metacarpophalangeal joints of 31 horses. The same parameters were used for sagittal, frontal and transverse sequences. T1W GRE—T1-weighted gradient echo, T2*W GRE—T2*-weighted gradient echo, T2W FSE—T2-weighted fast spin echo, STIR FSE—short tau inversion recovery fast spin echo, MI—motion insensitive, TE—echo time, ms—milliseconds, TR—repetition time, FE—frequency encoding, PE—phase encoding, FOV—field of view.

Pulse Sequence	TE (ms)	TR (ms)	Flip Angle (°)	Slice Thickness (mm)	Slice Gap (mm)	Matrix Size (FE × PE)	FOV (mm)
Pilot	7	66	45	7	0	150 × 120	220
Pilot of a pilot	7	66	45	7	0	150 × 120	220
STIR TEST	22	2100	50/60/85/110	4	0.8	256 × 144	200
T1W GRE MI	8	50	55	5	1	170 × 130	170
T2*W GRE MI	13	68	25	5	1	340 × 160	170
T2W FSE FAST	88	1544	90	5	1	168 × 168	170
STIR FSE FAST	22	2336	95	5	1	168 × 168	170

## Data Availability

Anonymised raw data are available upon reasonable request.

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
