# Peer review of "Magnetic Resonance Imaging, Computed Tomographic and Radiographic Findings in the Metacarpophalangeal Joints of 31 Warmblood Showjumpers in Full Work and Competing Regularly"

_animals, 2024, doi:10.3390/ani14101417_

Round 1
Reviewer 1 Report
Comments and Suggestions for Authors
Dear Authors,
Thank you for your submission and for the time you have taken putting together this prospective paper. I found your work very easy to read and I am very much looking forward to reading the results of your longitudinal work with this population of horses.
My only major question is why there was no more information on the presence or absence of STIR hyperintensity in this paper? If this is because there was little evidence of intraosseous fluid in these fetlocks then I believe it is important to draw attention to this, particularly in the fetlocks that had subchondral bone demineralisation. It will be very interesting to see if they develop regional fluid in your longitudinal data and I believe more attention could be given to the absence of fluid in your population of horses at this current time. I believe that the presence/absence of STIR hyperintensity and its clinical significance is quite controversial in the profession at this time and therefore really should be addressed at some point in your paper.
Other than that, I have a few minor comments as below:
Additional Comments:
Title: Can you please prefix showjumpers with “Warmblood” just to be clear that there are no off the track – re-purposed Thoroughbreds who may have come to your study with pre-existing fetlock pathology. Could you also please add this throughout the paper for consistency? (eg. Lines 301 and 451).
Line 122: If you consider all signal hypointensity as “densification”, how do you account for cases where intraosseous fluid could be contributing to the T1W hypointensity? Is this because you were reading the modalities simultaneously and could use the CT information to assist you? It appears you have some discrepancies between the MRI and CT finding with respect to densification so perhaps you need to address this issue here? Did you use the T2W FSE to assist with interpretation of what was true densification on MRI and what was fluid?
Line 125-128; Condylar densification is graded in many different ways in clinical practice and in the literature. Can you please expand upon how your grading system worked (eg. include an image?) You specify you used the transverse images on MRI for dorsopalmar length but were these the transverse images perpendicular to P1 or MC3? Your current description is unclear and would make it difficult for someone to repeat your methodoly/utilise your grading system in practice.
Could you also please consider adding your severity grades to the text and Figure legends when you are referring to densification so we see some examples of your grading scale in use?
Figures 6 and 7: Why do you think the modelling of the MCP joint was more visible with MRI than CT? This seems odd given the greater slice thickness on MRI. Is it possible we over-read modelling on MRI if it doesn’t exist on the corresponding CT images? Slice thickness/volume averaging could be contributing to this? Perhaps some additional explanation in your discussion (around Lines 424-426)?
Figure 7: How do you separate SCB thickening vs adjacent trabecular bone sclerosis. This is something I struggle with myself and I am interested in more explanation on this in your paper. Earlier in your paper you use “SCB thickening and/or densification of the trabecular bone” (Line 158) and I think it might be good to use this consistently throughout the paper (ie. here in Figure 7 and elsewhere) unless you have a way you distinguish between the 2? Surely the MRI hypointensity/CT hyperattenuation surrounding the focal area of demineralisation has to be too far away from the articular surface to still be considered SCB thickening? Or do you still consider that SCB?
Lines 350-353: I am also puzzled by why the palmar condylar densification was seen more frequently with CT than MRI – were these images definitely compared in the same imaging planes? It also confuses me as to why the conical densification identified dorsally was more prevalent on MRI than CT (ie. the opposite to the palmar condyles). Is it possible that there was mild intraosseous fluid associated with the dorsal sagittal ridge densification which would give the appearance of greater hypointensity (T1W)? This wouldn’t explain the disagreement between the condylar changes but might be a reason for the dorsal conical changes? Could you perhaps expand/explore this in the context of STIR hyperintensity?
Author Response
- Thank you for your submission and for the time you have taken putting together this prospective paper. I found your work very easy to read and I am very much looking forward to reading the results of your longitudinal work with this population of horses. My only major question is why there was no more information on the presence or absence of STIR hyperintensity in this paper? If this is because there was little evidence of intraosseous fluid in these fetlocks then I believe it is important to draw attention to this, particularly in the fetlocks that had subchondral bone demineralisation. It will be very interesting to see if they develop regional fluid in your longitudinal data and I believe more attention could be given to the absence of fluid in your population of horses at this current time. I believe that the presence/absence of STIR hyperintensity and its clinical significance is quite controversial in the profession at this time and therefore really should be addressed at some point in your paper.
A. Thank you for these comments. We apologise the text was not clear enough. STIR hyperintensity was only seen associated with focal demineralisation. ‘in all image sequences’ has been added to the text in lines 166 and 194.
We have also added a section in the Discussion on the lack of diffuse STIR hyperintensity in this population (lines 432-439).
- Title: Can you please prefix showjumpers with “Warmblood” just to be clear that there are no off the track – re-purposed Thoroughbreds who may have come to your study with pre-existing fetlock pathology. Could you also please add this throughout the paper for consistency? (eg. Lines 301 and 451).
A. ‘Warmblood’ has been added to the title and to lines previously numbered 301 and 451 for clarification (307 and 470 in the revised manuscript). In the Results (line 141) it is stated that ‘All 31 horses included in the study were Warmbloods’.
- Line 122: If you consider all signal hypointensity as “densification”, how do you account for cases where intraosseous fluid could be contributing to the T1W hypointensity? Is this because you were reading the modalities simultaneously and could use the CT information to assist you?
A. Thank you for this remark. Clarification has been added to the text (line 125):
‘concurrent hyperattenuation in CT reconstructions and hypointense signal in T1 and T2*-weighted gradient echo and T2-weighted fast spin echo MR images in the trabecular bone will be referred to as ‘densification’.’
- It appears you have some discrepancies between the MRI and CT finding with respect to densification so perhaps you need to address this issue here?
A. A sentence has been added to the limitations.
- Did you use the T2W FSE to assist with interpretation of what was true densification on MRI and what was fluid?
A. T2W FSE sequences were only acquired in the transverse plane. Yes, it was also used to confirm densification, but primarily, concurrent hypointense signal in both T1 and T2*-weighted GRE confirmed what was true densification.
- Line 125-128; Condylar densification is graded in many different ways in clinical practice and in the literature. Can you please expand upon how your grading system worked (eg. include an image?) You specify you used the transverse images on MRI for dorsopalmar length but were these the transverse images perpendicular to P1 or MC3? Your current description is unclear and would make it difficult for someone to repeat your methodoly/utilise your grading system in practice.
A. With regards to MRI, it is described in lines 92-93: ‘Transverse plane sequences were set perpendicular to the long axis of the proximal phalanx.’
CT images were reconstructed in multiple ways to allow the best evaluation of each structure, and this was usually perpendicular to the long axis of each bone or to the subchondral bone in case subchondral abnormalities were assessed. As the position of the limb is different in the low-field Hallmarq MRI (weight bearing, fetlock in extension) to the Qalibra CT system (limb extended forward, fetlock flexed to variable degree), it is impossible to reconstruct fully matching slices. We fully acknowledge that this is a shortcoming of the subjective assessment and would like to emphasise again that grading was only used in this study to support subjective assessment.
Further studies are under way that include detailed objective assessment (measurement of subchondral bone thickness, measurement of bone density based on Hounsfield units in the trabecular bone in numerous locations) and comparison of imaging modalities. For this reason, we do not think that additional figures would be useful in this paper. We believe the descriptions are now clear and the limitations have been acknowledged.
- Could you also please consider adding your severity grades to the text and Figure legends when you are referring to densification so we see some examples of your grading scale in use?
A. This was already included for most figures and has now been added to the remaining ones, where applicable.
- Figures 6 and 7: Why do you think the modelling of the MCP joint was more visible with MRI than CT? This seems odd given the greater slice thickness on MRI. Is it possible we over-read modelling on MRI if it doesn’t exist on the corresponding CT images? Slice thickness/volume averaging could be contributing to this? Perhaps some additional explanation in your discussion (around Lines 424-426)?
A. We agree that this is interesting, but we would prefer not to draw conclusions based on the small number of horses. In fact, periarticular modelling was seen in more limbs in CT than MR images (lines 240-245):
‘Periarticular modelling of the proximal phalanx was identified in 26/62 (41.9%) limbs in CT images (on both the medial and lateral aspects in 23/26 limbs and on the lateral aspect only in 3 limbs);and in 11/62 (17.7%) limbs in MR images (on both medial and lateral aspects in 4 limbs, on the medial aspect only in 1 limb, on the lateral aspect only in 6 limbs (Figures 2, 7)’.
We agree with the Reviewer, in Figures 6 and 7 the more obvious osteophytes on MRI can be due to volume averaging.
- Figure 7: How do you separate SCB thickening vs adjacent trabecular bone sclerosis. This is something I struggle with myself and I am interested in more explanation on this in your paper. Earlier in your paper you use “SCB thickening and/or densification of the trabecular bone” (Line 158) and I think it might be good to use this consistently throughout the paper (ie. here in Figure 7 and elsewhere) unless you have a way you distinguish between the 2?
A. We agree with the Reviewer, and we struggle too! In many locations we do not feel we can reliably differentiate. We have added a sentence to the Discussion (lines 348-350):
‘The authors acknowledge that in some horses differentiation between thickening of the subchondral bone and densification of the adjacent trabecular bone can be challenging.’
- Surely the MRI hypointensity/CT hyperattenuation surrounding the focal area of demineralisation has to be too far away from the articular surface to still be considered SCB thickening? Or do you still consider that SCB?
A. We do not think we know what came first, focal thickening or focal demineralisation followed by focal densification in the underlying subchondral/ trabecular bone. We share the Reviewers’ concerns about terminology and have amended the figure legend accordingly, adding ‘and/or densification of the trabecular bone’.
- Lines 350-353: I am also puzzled by why the palmar condylar densification was seen more frequently with CT than MRI – were these images definitely compared in the same imaging planes?
A. Yes the images were compared in the same imaging plane, and additional CT images, reconstructed perpendicular to the axis of the third metacarpal bone were also assessed.
- It also confuses me as to why the conical densification identified dorsally was more prevalent on MRI than CT (ie. the opposite to the palmar condyles). Is it possible that there was mild intraosseous fluid associated with the dorsal sagittal ridge densification which would give the appearance of greater hypointensity (T1W)? This wouldn’t explain the disagreement between the condylar changes but might be a reason for the dorsal conical changes? Could you perhaps expand/explore this in the context of STIR hyperintensity?
A. Please see previous comments (1. and 3., above). When describing densification, we were confident (as far as one can be with low-field MRI in a standing horse) that there was no intraosseous fluid, based on the hypointense signal in T2-weighted sequences and lack of hyperintense signal in STIR images.
Reviewer 2 Report
Comments and Suggestions for Authors
The presented article is a descriptive study on normal findings detected in the front fetlocks of showjumpers without relevant lameness examined with MRI, CT and radiography. No such publication on the fetlock is available at the moment. The authors presented the results and their discussion clearly and orderly.
General comments:
Unfortunately, the number of cases is limited, and no statistical test were performed as described as the main limitation of the study. In this study subjective and objective non lame horses and horses without relevant lameness were included. It would have been interesting to differentiate the imaging findings between the none lame and without relevant lameness cases, as this study is describing the normal imaging findings in healthy front fetlocks, but we should consider the cases without relevant lameness pre-stage to a lameness case.
At the end of the M&M section it is mentioned that no measurements were acquired, and that objective image analysis will be performed in a separate study. Why are these data not presented in this study, as this data can already be available?
Specific comments:
Line 59-62: please specify the choice in different terminology between non-lame and without relevant lameness.
Line 69-73: Were any imaging examinations performed prior to the study, please include grounds why you would have rejected a case for the study.
Line 142-156: describe the lameness diagnosis of the cases included in the study, were no diagnostic analgesia performed? These cases are not specifically reviewed based on the imaging findings described later?
Line 318-322: this studies describes the normal findings detected in front fetlocks, in some horses however the vector sum of head motion asymmetry was higher than the threshold for lameness detection recommended for this objective lameness assessment tool. Do you disagree with this cutoff value used and included these horse therefore in this study?
Author Response
The presented article is a descriptive study on normal findings detected in the front fetlocks of showjumpers without relevant lameness examined with MRI, CT and radiography. No such publication on the fetlock is available at the moment. The authors presented the results and their discussion clearly and orderly.
- Unfortunately, the number of cases is limited, and no statistical test were performed as described as the main limitation of the study. In this study subjective and objective non lame horses and horses without relevant lameness were included. It would have been interesting to differentiate the imaging findings between the none lame and without relevant lameness cases, as this study is describing the normal imaging findings in healthy front fetlocks, but we should consider the cases without relevant lameness pre-stage to a lameness case.
A. Thank you for the comment. We do not feel that the population could be split reliably into smaller groups based on a marginal gait abnormality presented on a single occasion. Moreover, in our experience, horses scan show a great variation in subtle and less subtle gait abnormalities when assessed on subsequent occasions. While some horses with subtle gait abnormality may progress and show overt lameness, others remain unchanged and perform successfully for years or the previously detected gait abnormality never appears again. To our knowledge this is the most comprehensive description of clinical examination and gait assessment findings in a study that investigated diagnostic imaging findings in horses considered not lame by their rider and trainer. We were completely transparent in reporting our observations; we are absolutely confident that had these horses been undergoing a pre-event gait assessment that all would have been considered fit to compete and a majority of observers would have considered that all of the horses were subjectively non-lame.
- At the end of the M&M section it is mentioned that no measurements were acquired, and that objective image analysis will be performed in a separate study. Why are these data not presented in this study, as this data can already be available?
A. Grading was used in this study to support the subjective assessment, for example the extent of trabecular bone densification. Further studies are under way that include detailed objective assessment (measurement of subchondral bone thickness, determination of bone density by measuring Hounsfield units in the trabecular bone in numerous locations) and comparison of imaging modalities. These are large amount of data and will be analysed and presented separately. The volume of data would have been too much to include in a single paper and would potentially have diluted the important subjective assessments.
- Line 59-62: please specify the choice in different terminology between non-lame and without relevant lameness.
A. Please see our previous comments (1, above). In previous diagnostic imaging studies, the term ‘non-lame’ was used rather liberally, often with no information provided on the lameness and clinical examinations. To us, non-lame is defined by a horse that does not show any lameness under any circumstances. Many actively performing horses show a subtle inconsistent gait abnormality under one or more circumstance that often does not impact their performance and this is what we mean by ‘without relevant lameness’. We have been honest and transparent in describing our observations. Please also bear in mind that horses were not just evaluated moving in hand in straight lines, but also on the lunge on soft and firm surfaces, so our inclusion criteria were stringent.
- Line 69-73: Were any imaging examinations performed prior to the study, please include grounds why you would have rejected a case for the study.
A. Imaging had not been performed consistently. Horses with previous joint medication, arthroscopy or any known metacarpophalangeal joint disease were not allowed to participate. This of course could not be ruled out for many horses under previous ownership and is acknowledged in the Discussion, lines 453-455.
‘The main limitation of the study is that although owners declared no knowledge of fetlock region problems, previous metacarpophalangeal joint disease (e.g., in previous ownership) cannot be excluded in all horses.’
A horse with a consistent lameness >grade 1/8 (grade 1 = just perceptible) under any circumstance (in hand in straight lines, or on the lunge on soft or hard surfaces) would have been excluded. This statement has been added to the Materials and Methods for clarification.
- Line 142-156: describe the lameness diagnosis of the cases included in the study, were no diagnostic analgesia performed?
A. The lameness observed was not consistent enough to allow reliable judgement of the effects of diagnostic anaesthesia. Had a horse shown a consistent lameness >grade 1/8 (grade 1 = just perceptible) under any circumstance (in hand in straight lines, or on the lunge on soft or firm surfaces) it would have been excluded from the study. On follow-up assessments, when lameness was more consistently apparent and > grade 1/8 in several horses, it was investigated using diagnostic anaesthesia.
6. These cases are not specifically reviewed based on the imaging findings described later?
A. Given the subtle and inconsistent nature of the lameness seen in a minority of horses, the absence of definitive localising clinical signs, the absence of verification of the source of pain causing lameness, and the small study population we think it would be potentially misleading to present the imaging findings of the non-lame horses and those in which the lameness was considered to be not clinically relevant, separately.
- Line 318-322: this studies describes the normal findings detected in front fetlocks, in some horses however the vector sum of head motion asymmetry was higher than the threshold for lameness detection recommended for this objective lameness assessment tool. Do you disagree with this cutoff value used and included these horse therefore in this study?
A. This is discussed in lines 321-327:
‘In the current study, the vector sum of head motion asymmetry was higher in some horses than the threshold for lameness detection recommended for this objective lameness assessment tool using body mounted inertial sensors (8.5 mm) [39]. Head motion asymmetry exceeding the published threshold level has been documented in event horses competing at 3* and 4* levels [38] and in young Warmblood horses in full work [40]. Mild movement asymmetry may be related to laterality [39-41], rather than pain-induced lameness.’
Round 2
Reviewer 1 Report
Comments and Suggestions for Authors
I am satisfied with the edits and comments provided by the authors. Thank you for your time and effort with this manuscript.